# Scavenging of Cation Radicals of the Visual Cycle Retinoids by Lutein, Zeaxanthin, Taurine, and Melanin

**DOI:** 10.3390/ijms25010506

**Published:** 2023-12-29

**Authors:** Malgorzata Rozanowska, Ruth Edge, Edward J. Land, Suppiah Navaratnam, Tadeusz Sarna, T. George Truscott

**Affiliations:** 1Cardiff Institute of Tissue Engineering and Repair, Cardiff University, Cardiff CF10 3AX, UK; 2School of Optometry and Vision Sciences, Cardiff University, Cardiff CF24 4HQ, UK; 3Dalton Cumbrian Facility, The University of Manchester, Westlakes Science Park, Moor Row, Cumbria CA24 3HA, UK; ruth.edge@manchester.ac.uk; 4The Paterson Institute, The University of Manchester, Wilmslow Road, Manchester M20 4BX, UK; e.land@mighty-micro.co.uk; 5Biomedical Sciences Research Institute, University of Salford, Manchester M5 4WT, UK; navaratnam1000@gmail.com; 6Department of Biophysics, Faculty of Biochemistry, Biophysics and Biotechnology, Jagiellonian University, 30-387 Krakow, Poland; tadeusz.sarna@uj.edu.pl; 7School of Chemical and Physical Sciences, Lennard-Jones Building, Keele University, Staffordshire ST5 5BG, UK; t.g.truscott@keele.ac.uk

**Keywords:** retinal, vitamin A, free radicals, xanthophylls, carotenoids, vitamin E, vitamin C, retina, age-related macular degeneration, Stargardt’s disease

## Abstract

In the retina, retinoids involved in vision are under constant threat of oxidation, and their oxidation products exhibit deleterious properties. Using pulse radiolysis, this study determined that the bimolecular rate constants of scavenging cation radicals of retinoids by taurine are smaller than 2 × 10^7^ M^−1^s^−1^ whereas lutein scavenges cation radicals of all three retinoids with the bimolecular rate constants approach the diffusion-controlled limits, while zeaxanthin is only 1.4–1.6-fold less effective. Despite that lutein exhibits greater scavenging rate constants of retinoid cation radicals than other antioxidants, the greater concentrations of ascorbate in the retina suggest that ascorbate may be the main protectant of all visual cycle retinoids from oxidative degradation, while α-tocopherol may play a substantial role in the protection of retinaldehyde but is relatively inefficient in the protection of retinol or retinyl palmitate. While the protection of retinoids by lutein and zeaxanthin appears inefficient in the retinal periphery, it can be quite substantial in the macula. Although the determined rate constants of scavenging the cation radicals of retinol and retinaldehyde by dopa-melanin are relatively small, the high concentration of melanin in the RPE melanosomes suggests they can be scavenged if they are in proximity to melanin-containing pigment granules.

## 1. Introduction

Upon absorption of light by the visual pigment in photoreceptive neurons of the vertebrate retina, its chromophore-11-*cis*-retinaldehyde is isomerized to all-*trans* configuration, which is followed by its hydrolysis from opsin and transient accumulation in photoreceptor outer segments (POS) until it is enzymatically reduced to all-*trans*-retinol [1,2]. To enable the regeneration of visual pigment, so it can absorb another photon of visible light to initiate a cascade of events leading to visual perception, the apoprotein needs to bind to another 11-*cis*-retinaldehyde, which needs to be delivered to photoreceptors mostly from the neighbouring retinal pigment epithelium (RPE). All-*trans*-retinol formed in POS is transported by chaperone proteins to the RPE [1,2,3,4]. In the RPE, all-*trans*-retinol, delivered either from photoreceptors or from blood, is the substrate for enzymes responsible for the synthesis of 11-*cis*-retinaldehyde. Firstly, it is esterified with fatty acids forming mainly all-*trans*-retinyl palmitate. All-*trans*-retinyl palmitate is used for retinoid storage or as a substrate of isomerohydrolase RPE65. All-*trans*-retinyl esters are stored in specialized lipid droplets called retisomes, and to enable them to become a substrate for RPE65 they need to be transferred to the endoplasmic reticulum (ER). It has been recently shown that patatin-like phospholipase domain containing 2 (PNPLA2) hydrolyses these esters to release all-*trans*-retinol so it can be transported to ER where it is esterified again by lecithin-retinol acyltransferase (LRAT) to become the substrate of RPE65 [5]. RPE65 converts all-*trans*-retinyl esters to 11-*cis*-retinol. Upon enzymatic oxidation of 11-*cis*-retinol, 11-*cis*-retinaldehyde is formed and is transported back to photoreceptive neurons by cellular and interphotoreceptor retinoid-binding proteins to regenerate the visual pigment and complete the major pathway of visual pigment chromophore synthesis in the visual cycle. Another pathway leading to regeneration of the visual pigment chromophore operates under exposure to light. Both RPE and Müller cells express retinal G-protein-coupled receptor (RGR) photoisomerase, which binds all-*trans*-retinaldehyde via Schiff-base linkage [6,7,8]. Upon absorption of light by RGR, all-*trans*-retinaldehyde is isomerized to 11-*cis* form, hydrolyzed from RGR, and transported by chaperone proteins to photoreceptors for regeneration of visual pigments.

In the retina, the trafficking of the visual cycle retinoids exposes them to the constant threat of oxidation due to the presence of potent photosensitizers exposed to light and under high oxygen tension and high concentration of polyunsaturated fatty acids susceptible to oxidation and propagation of lipid peroxidation via peroxyl radicals [9,10]. Lipid peroxidation can be facilitated in retinal diseases such as age-related macular degeneration (AMD), where RPE exhibits about a five-fold increased content of total iron and iron chelateable by desferrioxamine in comparison with RPE in age-matched normal retinas [11]. Increased levels of products of lipid peroxidation have been identified post-mortem in the human retinas affected by AMD, which is the predominant cause of vision loss in people above 50 years of age in developed countries [12,13,14,15,16,17]. It has been previously shown that retinoids, including retinol and retinaldehyde, can scavenge peroxyl radicals, thereby contributing to the antioxidant action of retinoids in lipid peroxidation where their antioxidant action can exceed that of vitamin E, a well-known chain-breaker in lipid peroxidation [18,19]. Scavenging of peroxyl radicals by retinoids can result in the formation of retinoid cation radicals. Moreover, retinoids can form cation radicals as a result of interaction with hydroxyl radicals or photoexcitation [20,21,22]. While retinaldehyde can be photoexcited by UV and blue light, the absorption spectra of retinol and retinyl palmitate extend very little into the visible range of light but show a relatively strong absorption at 320 nm. While most UV light is absorbed by the cornea and the lens of the adult human eye, the transmission window with a maximum at about 320 nm persists even in people above the age of 60 years [23].

Retinoid cation radicals can be damaging to amino acids and proteins [22]. For example, retinol cation radicals can oxidize several amino acids, such as cysteine, tryptophan, lysine, and arginine [22]. The formation of retinoid cation radicals can lead to their further degradation. It has been previously shown that a mixture of oxidation products of retinaldehyde retains the photosensitizing properties of retinaldehyde while becoming more (photo)toxic than the parent compound [24]. (Photo)toxicity of retinaldehyde and its degradation products is of particular relevance to blinding diseases caused by delayed clearance of all-*trans*-retinaldehyde, such as Stargardt’s disease caused by dysfunction of enzymes responsible for all-*trans*-retinaldehyde removal: ATP-binding cassette transporter A4 (ABCA4) or retinol dehydrogenase 8 (RDH8) [25,26,27]. Stargardt’s disease is usually diagnosed in childhood or early adulthood and progresses rapidly to severe loss of vision.

It has been previously shown that the cation radicals of retinol and retinaldehyde can be scavenged by vitamin E and vitamin C, presumably restoring the parent compound [18,28]. Vitamins C (ascorbate) and vitamin E, present mainly as α-tocopherol, are essential components of the vertebrate retina as well as of an antioxidant supplement which, together with zinc, lutein, and zeaxanthin, has been shown to decrease the risk of progression of a moderate form of AMD to its advanced form over a 10-year period from 49.2% to 47.3%, and the effect was statistically significant [29,30,31,32,33].

The potential formation of cation radicals of retinyl palmitate as a result of interaction with oxidizing radicals and the effect of vitamins C and E on retinyl palmitate cation radicals have not been investigated yet. In addition to vitamins E and C, the area where the trafficking of the visual cycle retinoids takes place contains other antioxidants. The POS and RPE of the human retina accumulate two carotenoids: lutein and zeaxanthin [34,35,36,37], which are thought to play important antioxidant functions in the retina [38,39]. The retinas of vertebrates contain high concentrations of taurine (2-amino-ethanesulfonic acid), which is essential for the viability of retinal neurons, including photoreceptors [40,41,42]. RPE contains eumelanin enclosed within melanosomes. which, especially in the young eye, appear mainly in the apical processes of the RPE extending into the layer of POS (reviewed in [43]). With ageing, RPE accumulates other melanin-containing granules—melanolysosomes and melanolipofuscin. Therefore, the aim of this study was to investigate the formation of retinyl palmitate cation radicals as a result of electron transfer from oxidizing radicals and the interactions of cation radicals of all three retinoids involved in the visual cycle with the major antioxidants present in the retina: ascorbate, lutein, zeaxanthin, taurine, and melanin.

The retinoid radical cations were generated by pulse radiolysis of nitrous oxide-saturated benzene or of aqueous buffered solution of potassium bromide, where retinoids were solubilized in Triton X-100 micelles. We observed retinyl palmitate cation radicals formation as a result of the interaction of retinyl palmitate with both dibromine anion radicals and benzene cation radicals. As a result of interaction with dibromine anion radicals, retinyl palmitate in micelles formed its cation radical with an absorption maximum at 590 nm, whereas in benzene, an electron transfer from retinyl palmitate to cation radicals of benzene resulted in the formation of retinyl palmitate cation radicals with an absorption maximum at 610 nm. Retinyl palmitate cation radicals were scavenged by α-tocopherol and by ascorbate so the bimolecular rate constants of scavenging could be determined. We observed no effect of taurine on the rate of cation radical decay for any of the retinoids. Synthetic dopa-melanin, used as a model of eumelanin present in the RPE, had no effect on the rate of decay of cation radicals of retinyl palmitate but scavenged cation radicals of retinol and retinaldehyde. Lutein and zeaxanthin scavenged cation radicals of all three retinoids with the bimolecular rate constants approaching the diffusion-controlled limits in benzene. The physiological relevance of the results is discussed in terms of concentrations of antioxidants in the retina. Despite that lutein and zeaxanthin tend to be more effective scavengers of retinoid cation radicals than ascorbate, the greater concentrations of ascorbate in the retina suggest that vitamin C may be the main protectant of visual cycle retinoids from oxidation. Lutein and zeaxanthin are likely to offer substantial contribution to retinoid cation radical scavenging oxidation only in the macula.

## 2. Results

### 2.1. Formation of Cation Radical of Retinyl Palmitate in Benzene

The pulse radiolysis of N_2_O-saturated benzene in the presence of 1 mM of retinyl palmitate resulted in the formation of retinyl palmitate triplet state with an absorption maximum at about 410 nm and retinyl palmitate cation radicals exhibiting an absorption maximum at 610 nm and decaying with kinetics, which, after omitting the initial contribution to the decay from the triplet state, could be fitted with a single exponential decay (Figure 1A,B).

### 2.2. Interaction of Cation Radical of Retinyl Palmitate with α-Tocopherol, Lutein, and Zeaxanthin

In the presence of α-tocopherol, the decays of retinyl palmitate cation radicals were faster than in its absence, allowing for the determination of the bimolecular rate constant of scavenging as (2.7 ± 0.3) × 10^7^ M^−1^s^−1^ (Figure 1A,C; Table 1).

In the presence of zeaxanthin, due to the high absorption coefficient of the zeaxanthin ground state, the changes in the transmittance changes could not be monitored below 520 nm wavelength (Figure 1D; Table 1). As expected, a formation of a zeaxanthin triplet state with an absorption maximum at about 510–520 nm could be seen, where the majority of the zeaxanthin triplet state appeared to be a result of energy transfer from the triplet state of retinyl palmitate (Figure 1D) [44,45]. The absorption of the zeaxanthin triplet state partly overlapped with the absorption of the retinyl palmitate cation radical. In the presence of zeaxanthin, the lifetime of retinyl palmitate cation radical was substantially shortened, and its decay was accompanied by the concomitant formation of zeaxanthin cation radical with an absorption maximum at about 1000 nm [46] (Figure 1D–F). The bimolecular rate constant of scavenging of retinyl palmitate cation radicals by zeaxanthin was (8.80 ± 0.05) × 10^9^ M^−1^s^−1^ when determined based on rates of decay at 610 nm and (6.39 ± 0.02) × 10^9^ M^−1^s^−1^ when determined based on rates of formation of zeaxanthin cation radical at 1000 nm (Figure 1D–F; Table 1). The apparent faster bimolecular rate constant of scavenging of retinyl palmitate cation radicals based on 610 nm data suggested that the zeaxanthin triplet state could affect the kinetics at 610 nm, and therefore we took the value obtained from the rates of formation of the zeaxanthin cation radical as a more accurate value (Table 1).

In the presence of lutein, the lifetime of retinyl palmitate cation radical was also substantially shortened, and its decay was accompanied by the concomitant formation of lutein cation radical with an absorption maximum at about 950 nm [46] (Figure 1D–F). The bimolecular rate constants of scavenging of retinyl palmitate cation radicals by lutein were (8.97 ± 0.05) × 10^9^ M^−1^s^−1^ and (8.85 ± 0.25) × 10^9^ M^−1^s^−1^ when determined based on rates of decay at 610 nm and rates of formation of at 950 nm, respectively (Figure 1D–F; Table 1). While both values appear to be the same within the experimental uncertainty, for consistency we used the value based on rates of the formation of lutein cation radical (Table 1).

### 2.3. Interaction of Retinyl Palmitate with Dibromine Radical Anions

To create retinyl palmitate cation radical in Triton X-100 micelles, we used pulse radiolysis to create highly oxidizing dibromine radical, which we have shown previously abstracts electrons from retinol and retinaldehyde thereby creating cation radicals of these retinoids with absorption maxima at 590 nm. In case of retinyl palmitate, there was also a formation of a transient species with the absorption maximum at 590 nm (Figure 2A). In addition, there was another species formed with an absorption maximum at about 400 nm. Some of this species was formed within the electron pulse with a subsequent growth, which, judging by an isosbestic point at about 460 nm, was due to the decomposition of the radical cation of retinyl palmitate. Drawing analogy from data on carotenoids presented by Polyakov et al., it can be suggested that the transient at 400 nm is a neutral radical of retinyl palmitate formed as a result of proton loss from the cation radical [47].

### 2.4. Interaction of Cation Radical of Retinyl Palmitate with Ascorbate, Taurine, and Dopa-Melanin

In the presence of ascorbate, the rates of decay of retinyl palmitate cation radical were increased, and the calculated bimolecular rate constant of scavenging of retinyl palmitate cation radicals was (5.8 ± 0.1) × 10^8^ M^−1^s^−1^ (Figure 2; Table 1).

Taurine or dopa-melanin had no effect on the rate of decay of the retinyl palmitate cation radical, suggesting that if the scavenging occurs, the bimolecular rate constant is smaller than 2 × 10^7^ M^−1^s^−1^ (Figure 3; Table 1).

### 2.5. Interaction of Cation Radical of Retinaldehyde with Lutein, and Zeaxanthin

Similar to the previous findings, pulse radiolysis of N_2_O-saturated benzene in the presence of 1 mM retinaldehyde resulted in the formation of its triplet state species and cation radical with absorption maximum at 610 nm [28,45]. While the triplet state decayed faster than the cation radical, it did contribute to the decay kinetics observed at 610 nm, and therefore we omitted that initial part from the fitting of the exponential decay to obtain a more accurate value of the rate of decays of the retinaldehyde cation radical (Figure 4A). In the presence of lutein, the lifetime of retinaldehyde cation radical was substantially shortened, and its decay was accompanied by the concomitant formation of lutein cation radical with an absorption maximum at about 950 nm (Figure 4A–D). The bimolecular rate constants of scavenging of retinaldehyde cation radicals by lutein were (1.10 ± 0.02) × 10^10^ M^−1^s^−1^ and (1.15 ± 0.14) × 10^10^ M^−1^s^−1^ when determined based on rates of decay at 610 nm and rates of formation of at 950 nm, respectively (Table 1). In the presence of zeaxanthin, the lifetime of retinaldehyde cation radical was also substantially shortened, and its decay was accompanied by the concomitant formation of zeaxanthin cation radical with an absorption maximum at about 1000 nm (Figure 4E,F). The bimolecular rate constants of scavenging of retinaldehyde cation radicals by zeaxanthin were (7.63 ± 0.55) × 10^9^ M^−1^s^−1^ and (6.48 ± 0.29) × 10^9^ M^−1^s^−1^ when determined based on rates of decay at 610 nm and rates of formation of at 950 nm, respectively.

### 2.6. Interaction of Cation Radical of Retinaldehyde with Taurine, and Dopa-Melanin

Taurine had no effect on the rate of decay of the retinaldehyde cation radical, suggesting that, if the scavenging occurs, the bimolecular rate constant is smaller than 1 × 10^7^ M^−1^s^−1^ (Figure 5; Table 1). In the presence of dopa-melanin, the lifetime of retinaldehyde cation radical was shortened and the bimolecular rate constant of scavenging was (1.6 ± 0.8) × 10^7^ M^−1^s^−1^ (Figure 5; Table 1).

### 2.7. Interaction of Cation Radical of Retinol with Lutein, and Zeaxanthin

Pulse radiolysis of N_2_O-saturated benzene in the presence of 1 mM retinol resulted in the formation of transient species similar to that observed before [28,45]. In the presence of lutein, the lifetime of retinol cation radical was substantially shortened, and its decay was accompanied by the concomitant formation of lutein cation radical with an absorption maximum at about 950 nm (Figure 6A–C). The bimolecular rate constants of scavenging of retinaldehyde cation radicals by lutein were (1.01 ± 0.05) × 10^10^ M^−1^s^−1^ and (1.26 ± 0.04) × 10^10^ M^−1^s^−1^ when determined based on rates of decay at 610 nm and rates of formation of at 950 nm, respectively (Table 1). In the presence of zeaxanthin, the lifetime of retinol cation radical was also substantially shortened, and its decay was accompanied by the concomitant formation of zeaxanthin cation radical with an absorption maximum at about 1000 nm (Figure 6D,E). The bimolecular rate constants of scavenging of retinaldehyde cation radicals by zeaxanthin were (10.3 ± 0.6) × 10^10^ M^−1^s^−1^ and (7.9 ± 0.3) × 10^9^ M^−1^s^−1^ when determined based on rates of decay at 610 nm and rates of formation of at 950 nm, respectively. These scavenging rates are somewhat greater than the rate of scavenging of retinol cation radicals by zeaxanthin in methanol of (5.8 ± 0.8) × 10^9^ M^−1^s^−1^ [21].

### 2.8. Interaction of Cation Radical of Retinol with Taurine, and Dopa-Melanin

Like for other retinoids, taurine had no effect on the rate of decay of the retinol cation radical, suggesting that, if the scavenging occurs, the bimolecular rate constant is smaller than 2 × 10^6^ M^−1^s^−1^ (Figure 7; Table 1). In the presence of dopa-melanin, the lifetime of retinol cation radical was shortened and the bimolecular rate constant of scavenging was (5.1 ± 0.1) × 10^6^ M^−1^s^−1^ (Figure 5; Table 1).

## 3. Discussion

### 3.1. Scavenging of Retinyl Palmitate Cation Radicals by Retinal Antioxidants

We have determined that pulse radiolysis of N_2_O-saturated benzene in the presence of retinyl palmitate leads to the formation of retinyl palmitate cation radicals with absorption maxima at 610 nm, whereas pulse radiolysis of buffered aqueous solutions of KBr in the presence of retinyl palmitate solubilized in Triton-X micelles leads to the formation of retinyl palmitate cation radicals with absorption maxima at 590 nm. Under the experimental conditions used, cation radicals of retinyl palmitate are not scavenged by melanin or taurine, which allowed us to determine that if they do scavenge retinyl palmitate cation radicals, the upper limits of the bimolecular rate constants are smaller than 2 × 10^7^ M^−1^s^−1^. The cation radicals of retinyl palmitate are scavenged by lutein, zeaxanthin, α-tocopherol, and ascorbate with bimolecular rate constants of (8.85 ± 0.25) × 10^9^ M^−1^s^−1^, (6.39 ± 0.02) × 10^9^ M^−1^s^−1^, (2.7 ± 0.3) × 10^7^ M^−1^s^−1^ and (5.8 ± 1.0) × 10^8^ M^−1^s^−1^, respectively. Interestingly, α-tocopherol appears to be less efficient than both carotenoids in scavenging retinyl palmitate cation radicals. To evaluate the relative contributions of ascorbate, α-tocopherol, lutein, and zeaxanthin to scavenging of retinyl palmitate cation radicals in the retina, the concentrations of these antioxidants need to be considered in the RPE where retinyl palmitate accumulates.

Sommerburg et al. determined the concentration of lutein and zeaxanthin in macular and peripheral RPE from 11 cadavers. Lutein and zeaxanthin concentrations from the macular area of 6 mm in diameter are (0.27 ± 0.07) ng/mm^2^ and (0.18 ± 0.05) ng/mm^2^, respectively, whereas outside that area they are (7.05 ± 0.9) ng/tissue and (1.97 ± 0.33) ng/tissue, respectively [34]. Taking 13.63 cm^2^ as the total retinal surface [48] and 8 µm as the height of the RPE layer allows us to estimate the molar concentrations of these carotenoids: 33.8 and 22.5 µM for the average concentration of lutein and zeaxanthin in the macular area (6 mm in diameter) and 1.14 and 0.32 µM in the periphery.

The measurements of Rapp et al. [35] and Sommerburg et al. [34] were done before supplements of lutein and zeaxanthin became commercially available. Rapp et al. demonstrated that there is a linear positive correlation between the content of lutein and zeaxanthin in POS and the content of these carotenoids in the central macula of 3 mm in diameter, which is the area in the retina where lutein and zeaxanthin accumulate in the greatest concentration, mainly in the inner retinal layers [35,38,49]. It has been shown that the content of these carotenoids in the retina can be increased on average 2.9-fold by increased dietary intake and/or supplementation [50]. Therefore, it can be considered that the average concentrations of lutein and zeaxanthin in the RPE can also be increased 2.9-fold, achieving 98 and 65 µM for the average concentration of lutein and zeaxanthin in the macular area and 3.3 and 0.9 µM in the periphery. The highest concentrations of lutein and zeaxanthin reported in persons with high intake of these carotenoids were 5.1- and 5.9-fold greater for lutein and zeaxanthin, respectively, than the average values for people with normal intake. This gives us the estimate of the maximal concentrations of these carotenoids in the RPE: 172 and 133 µM for the maximal concentration of lutein and zeaxanthin in the macular area and 5.8 and 1.9 µM in the periphery.

The concentrations of lutein and zeaxanthin in different retinal areas allow evaluation of the effectiveness of their potential scavenging of retinoid cation radicals (Table 2). Table 2 shows the products of multiplications of antioxidant concentrations in different retinal areas/tissues and the bimolecular rate constant of retinyl palmitate cation radical scavenging. This allows comparison of the contribution of different antioxidants to such scavenging. In most cases, these data are based on the average concentration of antioxidants in the entire RPE cells. It can be seen that, without supplementation, lutein may be 2.1-fold more effective than zeaxanthin in scavenging retinyl palmitate cation radicals in the macular RPE and 5-fold more effective than zeaxanthin in the periphery. However, the overall effectiveness of both carotenoids strongly decreases in the periphery: 30 times for lutein and 71 times for zeaxanthin.

It is likely that antioxidants such as carotenoids have increased concentrations in certain subcellular compartments [36]. It has been shown that lutein can act as a substrate of the same enzyme, RPE65, which converts all-trans-retinyl palmitate into 11-*cis*-retinol [51]. In the case of lutein, RPE65 converts it into meso-zeaxanthin, which then accumulates in the neural retina, particularly in the macula. These data are based on studies on developing chicken eyes and cultured cells, expressing either human or chicken RPE65. The accumulation of meso-zeaxanthin in the chicken retina coincides with an increased expression of RPE65 and can be inhibited by an inhibitor of RPE65, which is also effective in inhibiting the visual pigment chromophore synthesis. These results, together with the well-documented accumulation of meso-zeaxanthin in the macula of the human retina, suggest that lutein and retinyl palmitate can colocalize in proximity to RPE65, which is present in the endoplasmic reticulum. Arguably, the retinoid cation radical may be involved in the combined reaction of isomerization and hydrolysis of all-trans-retinyl palmitate into 11-cis-retinol [52,53,54]. This raises a possibility, worth investigating, that lutein may act as an inhibitor of RPE65 activity towards retinyl palmitate not only as a competitive substrate but also as a scavenger of retinoid radical cation if it is involved in that process.

Another place of possible co-localization of lutein/zeaxanthin with retinyl palmitate is specialized lipid droplets, retinosomes, which are the major storage site of retinyl esters in the retina [1].

Friedrichson et al. have determined that the concentrations of vitamin E in the RPE are similar in the macula and periphery of the human retina [29]. Based on a similar approach as for carotenoids above, the calculated average concentration of vitamin E is 115 µM. This concentration exceeds almost four-fold the concentration of this vitamin in blood plasma (25–35 µM) [55,56]. Based on the highest content of vitamin E determined in that study being about twice greater than the average, the highest concentration of vitamin E was determined as 230 µM. As can be seen in Table 2, vitamin E may be a slightly more effective scavenger of retinyl palmitate cation radicals than zeaxanthin in the peripheral retina but not in the macula, where zeaxanthin is at least an order of magnitude more effective. The maximum concentration of vitamin E is not enough to provide similar scavenging as the average concentration of lutein in the periphery. In the macula, lutein has the potential to provide scavenging close to two orders of magnitude greater than α-tocopherol despite the average concentration of lutein is lower than that of vitamin E.

The concentration of ascorbate was determined in the vitreous as 2 mM [30], and, due to the lack of any barriers, it can be assumed to be similar in the interphotoreceptor matrix. This concentration hugely exceeds the typical concentrations of ascorbate in the blood plasma of 0.05 to 0.08 mM [56,57]. While, to our knowledge, there are no measurements of ascorbate in the human retinal neurons or RPE, the high expression of sodium-dependent vitamin C transporters in the basal plasma membrane of human RPE and, to a smaller extent, in photoreceptors and other cells of the neural retina suggests that RPE and retinal neurons can accumulate similar concentrations of ascorbate as glial cells and neurons in the brain: about 1 and 10 mM [30,58,59], respectively. Vitamin C, even at the lowest of these concentrations, appears to be able to provide the greatest contribution to scavenging of retinyl palmitate cation radicals in the peripheral RPE where its efficiency can be greater than that of maximal concentrations of lutein or vitamin E. In the macula, it may provide better scavenging than average concentrations of lutein and zeaxanthin combined.

### 3.2. Scavenging of Retinol Cation Radicals by Retinal Antioxidants

The cation radicals of retinol are scavenged by lutein and zeaxanthin more effectively than by α-tocopherol or ascorbate (Table 1). To evaluate their potential contributions to scavenging retinol cation radicals, we considered their concentrations in the retina (Table 3). It has been shown that about 15–25% of retinal carotenoids are present in POS [34,35]. Sommerburg et al. determined the content of lutein and zeaxanthin in POS isolated from the entire human retina so their molar concentrations can be estimated as 1.01 µM for lutein and 0.47 µM for zeaxanthin [34,37]. It is reasonable to expect that in the macula, where the concentrations of these carotenoids in the RPE are several-fold greater than in the periphery and their concentrations in the inner retina of the central macula reach millimolar values, the concentrations of lutein and zeaxanthin in POS will be also greater than in the periphery. Therefore, we have estimated these concentrations based on 2.1% contribution of the macular area of 6 mm in diameter to the total retinal surface of 13.63 cm^2^ and the ratio of respected carotenoid concentrations in the RPE of the macula and periphery. This approach gives 18.8 and 13.7 µM concentrations of lutein and zeaxanthin, respectively, in the macula and 0.63 and 0.18 µM in the periphery. It is likely that the concentrations are underestimated, possibly several-fold, due to losses of POS occurring in the isolation procedure. Table 3 shows that the potential effectiveness of retinol radical scavenging by lutein and zeaxanthin is the greatest in the RPE and POS in the macular area, and it drops at least an order of magnitude in the periphery.

It has been determined that the interphotoreceptor retinoid-binding protein (IRBP) binds lutein and zeaxanthin with similar affinity as retinoids [60]; therefore, it is likely that lutein and zeaxanthin are present in this area on their way to photoreceptors and the inner retina. To our knowledge, there are no reports on concentrations of lutein/zeaxanthin in the interphotoreceptor matrix and on how their binding to IRBP affects their interactions with free radicals; therefore, we cannot evaluate their potential contribution to scavenging of retinoid cation radicals in this area. We also have no such data for another lipophilic antioxidant, vitamin E, which is expected to be mostly solubilized in lipid membranes.

Based on data of Friedrichson et al., it has been determined that molar concentrations of vitamin E in the neural retina of the macular area and periphery are 46 and 77 µM, respectively [29,37]. Comparison of calculated effectiveness of retinol cation radicals scavenging clearly shows that, in comparison with lutein and zeaxanthin, α-tocopherol can provide only a minor contribution to the scavenging of the retinol cation radical in the macula but is similar or exceeds the contribution of these carotenoids in the periphery despite that the rate constant for scavenging of retinol cation radicals is 158 times greater for lutein than for α-tocopherol (Table 3).

Assuming 10 mM concentrations of ascorbate in photoreceptors, 2 mM in the interphotoreceptor matrix, and 1 mM in RPE, it can be estimated that, outside the macula, ascorbate can provide a greater contribution to scavenging of retinol cation radicals than any other antioxidant (Table 2). In the macula, ascorbate may provide a contribution to scavenging similar to that of lutein and zeaxanthin, both in POS and RPE. It can be expected that ascorbate can exert its antioxidant action at the interfaces of aqueous solution and lipid membranes as well as in the aqueous milieu. While it can be expected that, in the normal retina, the majority of retinol in the interphotoreceptor matrix is bound to IRBP [3], some may be solubilized directly in its aqueous part. It has been determined that the solubility of retinol in buffered water varies from 51 to 79 nM, depending on the evaluation method, and that ascorbate and α-tocopherol (in ethanolic/water solutions) can protect it from oxidative degradation in such environment [61]. Our results provide a potential mechanism that can be responsible for this protective effect and suggest that ascorbate could serve as the main protectant of retinol in the interphotoreceptor matrix.

Although the determined rate constant of the interaction of the cation radicals of retinol with synthetic dopa-melanin is relatively low, (5.1 ± 0.1) × 10^6^ M^−1^s^−1^, the obtained data do not rule out a possible protective role of melanin against oxidation by this radical in the RPE. In the retina, melanin is present in melanosomes and in melanolipofuscin granules (reviewed in [43]). Melanosomes, particularly in the young retina, are abundant in the apical processes of the RPE surrounding POS. Particularly these melanosomes are in the trafficking pathway of retinol diffusing, either as free retinol or bound by cellular retinol-binding protein (CRBP), toward the cell body where it becomes esterified. With ageing, melanosomes redistribute throughout the entire RPE cell while melanolysosomes and melanolipofuscin granules start to accumulate mainly in the cell body. It has been determined that melanin content in human RPE melanosomes isolated from 20–30-year-old cadavers is (0.18 ± 0.01) pg/melanosomes and decreases to (0.14 ± 0.2) pg/melanosomes for melanosomes isolated from 60–90-year-old cadavers [62]. Taking 2 µm as the long axis and 0.6 µm as the short axis of melanosome and calculating its volume as the volume of an ellipsoid and 150 g/mol as the molecular weight of melanin monomer, the concentration of melanin within melanosome can be estimated as 398 mM in young RPE and 310 mM in old RPE. Due to this high local concentration of melanin in RPE melanosomes and the strong affinity of melanin for positively charged molecules, it can be expected that retinol cation radicals formed in the proximity of RPE melanosomes will be effectively scavenged by melanin. It can be also expected that melanosomal concentrations of amino acids susceptible to oxidation by retinol cation radicals, namely, cysteine, tryptophan, lysine, and arginine, are orders of magnitude smaller than that of melanin. These amino acids scavenge retinol cation radicals with similar bimolecular rate constants as melanin, that is 2.6 × 10^5^, 7.1 × 10^5^, 1.4 × 10^7^, 1.5 × 10^7^, and 5.0 × 10^7^ M^−1^s^−1^ for cysteine, tryptophan, lysine, and arginine, respectively [22]. Therefore, it can be expected that, due to the high concentration of melanin within melanosomes, they can be effectively protected from oxidation by retinol cation radicals.

### 3.3. Scavenging of Retinaldehyde Cation Radicals by Retinal Antioxidants

The cation radicals of retinaldehyde are scavenged by lutein and zeaxanthin with similar bimolecular rate constants as that of α-tocopherol and about 16- and 9-fold greater, respectively, than by ascorbate (Table 1). In the case of scavenging of retinaldehyde cation radicals in the retina, α-tocopherol and ascorbate may provide a major contribution both in the RPE and neural retina (Table 4). Lutein, followed by zeaxanthin, may provide substantial protection in the RPE and POS in the macular area but only minor protection in the periphery.

The rate constant of scavenging of retinaldehyde cation radicals by dopa-melanin is about 3.1-fold greater than that for retinol cation radicals. Similar considerations as in the case of potential interactions of retinol cation radicals with melanin-containing pigment granules discussed above can be applied to the potential interactions of melanin with retinaldehyde cation radicals. After its release from the visual pigments, all-trans-retinaldehyde may escape its enzymatic reduction to all-trans-retinol in POS and diffuse to the apical processes of the RPE filled with melanosomes. Moreover, the tips of POS are phagocytosed daily by the RPE and enter its phagolysosomal system, where retinaldehydes are released as a result of proteolysis of visual pigments. In Stargardt’s disease caused by mutations resulting in the loss of function in ATP-binding cassette transporter subfamily A member 4 (ABCA4) [27,63], the removal of retinaldehydes from phagocytosed tips of POS can be further compromised, thereby increasing the risk of its oxidation. Such retinaldehydes may appear in proximity to melanolysosomes and melanolipofuscin. Also, the newly synthesized 11-cis-retinaldehyde may be in sufficient proximity to melanolysosomes and melanolipofuscin to facilitate its interaction with melanin once it becomes oxidized to cation radical.

Another enzyme responsible for the clearance of all-trans-retinaldehydes from POS is retinol dehydrogenase 8 (RDH8), which is the main enzyme there responsible for the reduction of all-trans-retinaldehyde to all-trans-retinol [1,64]. RDH8 dysfunction also causes Stargardt’s disease [26]. It can be expected that the delayed clearance of all-trans-retinaldehyde due to the loss of function of ABCA4 or RDH8 can affect mostly POS and subsequently the entire photoreceptor cell and RPE [64,65].

Retinaldehyde can accumulate in POS in substantial concentrations also in the healthy retina as a result of exposure of the dark-adapted retina to bright light that causes photoactivation of most visual pigments followed by the release of all-trans-retinaldehyde at concentrations up to 3.8 mM [1,64,65]. With ageing of the retina, there is an increased frequency of POS losing their ordered appearance and becoming convoluted and mismanaged [66]. This raises a possibility that retinoid trafficking and their enzymatic transformations can be affected and clearance of retinaldehydes delayed. Retinaldehydes are potent photosensitizers that upon absorption of UV or blue light photosensitize the generation of singlet oxygen, free radicals, and lipid peroxidation. POS contain a high concentration of polyunsaturated fatty acids, including docosahexaenoate with 6 unsaturated double bonds, making it extremely susceptible to oxidation by singlet oxygen or free radicals and subsequent propagation of lipid peroxidation, especially in the aged retina and retina affected by AMD, which exhibits increased levels of iron [9,10,11,67]. This oxidative environment, to which retinaldehydes contribute, can lead to the formation of retinaldehyde cation radicals. Moreover, further photooxidative degradation of retinaldehydes leads to the formation of products with increased cytotoxicity and phototoxicity in comparison to retinaldehyde [24]. Therefore, the adequate antioxidant protection of the retina is of utmost importance. Lutein, zeaxanthin, and vitamins E and C can provide antioxidant protection by quenching excited triplet states of retinaldehydes and singlet oxygen and by inhibiting lipid peroxidation [36,68,69]. The results of this study suggest that scavenging retinaldehyde cation radical may be another way these antioxidants can offer protection.

### 3.4. Summary and Wider Physiological Relevance

In summary, we have shown the formation of retinyl palmitate cation radicals both in benzene and in Triton X-100 micelles solubilized in phosphate buffer. Vitamin C scavenged retinyl palmitate radical cation with a bimolecular rate constant of (5.8 ± 0.1) × 10^8^ M^−1^s^−1^. This value is 4.8 times greater than the bimolecular rate constant of scavenging by ascorbate of cation radicals of retinol but 21% smaller than that for scavenging of cation radicals of retinaldehyde determined previously [18]. Vitamin E was 3- and 296-fold less effective in scavenging cation radicals of retinyl palmitate than in scavenging cation radicals of retinol or retinaldehyde, respectively.

We observed no effect of taurine, up to the highest concentration studied of 0.1 mM, on the rates of cation radical decays for any of the retinoids studied. Dopa-melanin had no effect on the decays of cation radicals of retinyl palmitate but did scavenge cation radicals of retinol and retinaldehyde with the bimolecular rate constants (5.1 ± 0.1) × 10^6^ M^−1^s^−1^ and (1.6 ± 0.8) × 10^7^ M^−1^s^−1^.

Lutein and zeaxanthin scavenged cation radicals of all three retinoids with the bimolecular rate constants approaching the diffusion-controlled limits in benzene, ranging from 6.4 × 10^9^ M^−1^s^−1^ for retinyl palmitate cation radical scavenged by zeaxanthin to 12.6 × 10^9^ M^−1^s^−1^ for retinol cation radical scavenged by lutein. Despite lutein exhibiting greater scavenging rate constants of retinoid cation radicals than other antioxidants, the greater concentrations of ascorbate in the retina suggest that vitamin C may be the main protectant of the visual cycle retinoids from oxidative degradation. Vitamin E may play a substantial role in the protection of retinaldehyde but may be relatively inefficient in the protection of retinol or retinyl palmitate. While the protection of retinoids by lutein and zeaxanthin appears inefficient in the retinal periphery, it can be quite substantial in the macula. It needs to be stressed that the calculations are based on carotenoid concentrations averaged from the entire macula. Lutein and zeaxanthin are concentrated in the centre of the macula of about 3 mm in diameter. Therefore, there is a possibility that the contributions of lutein and zeaxanthin in that area may be much greater than those estimated in Table 1, Table 2 and Table 3. Moreover, Bhosale et al. reported a great inter-individual variability in the contents of these carotenoids, which, in the macular area of 4 mm in diameter, range from about 1 ng up to about 115 ng [50]. This indicates that the inter-individual ability in scavenging of retinoid cation radicals by lutein and zeaxanthin can vary by a factor of over 100.

Although the determined rate constants of the interaction of the cation radicals of retinol and retinaldehyde with synthetic dopa-melanin are relatively low, the obtained data do not rule out a possible protective role of melanin in the RPE. This is because the local concentration of melanin in RPE melanosomes is very high, which would facilitate effective scavenging of the cation radicals by melanin if the cation radicals are formed in the proximity of RPE melanosomes.

While scavenging of retinoid cation radicals by lutein and zeaxanthin is a good thing because it prevents further oxidative degradation of retinoids and protects amino acids that otherwise could be oxidized by retinoid cation radicals, it is worth considering that the products of these reactions: cation radicals of lutein and zeaxanthin retain the ability to oxidize certain amino acids, namely tyrosine and cysteine [70]. However, the radical cations of these carotenoids can be reduced, thereby recycled to the parent molecules, by α-tocopherol, ascorbate, and melanin [70,71]. Therefore, it can be expected that these antioxidants working in combination with lutein and zeaxanthin can offer a synergistic protection. It has been previously shown that zeaxanthin in combination with α-tocopherol and/or ascorbate can often offer synergistic protection of unsaturated lipids or cultured cells exposed to light in the presence of photosensitizers [36,37,69,72,73,74]. The synergistic effect in protection against photooxidative damage, where both singlet oxygen and free radicals were generated, was ascribed to the ability of α-tocopherol and/or ascorbate to protect zeaxanthin from its oxidative degradation, thereby functioning for longer as an unsurpassed singlet oxygen quencher. This effect was particularly striking in a study on cultured ARPE-19 cells exposed to visible light and retinaldehyde-containing liposomes as a model of POS, where single antioxidants offered very small protective effects but their combinations provided a substantial synergistic effect [37]. It can be argued that scavenging of cation radicals of retinaldehyde could contribute to that protective effect.

The main limitation of this study is the extrapolation of the results obtained in a homogenous solution and Triton-X micelles to the retina and using for calculations the antioxidant concentrations that were averaged over the entire RPE cells or, in the case of vitamin E, over the entire neural retina. Benzene was meant to mimic the non-polar environment of the lipid membrane. However, the real retinal lipid membranes are complex structures with polar groups of retinol, retinaldehyde, lutein, and zeaxanthin likely to face the lipid-aqueous interface. Gao et al. reported that the polarity of the environment can exert substantial effects on the oxidation potentials of retinol and β-carotene and two other carotenoids [75]. Therefore, it would be of great interest to determine the dependence of the oxidation potential on the solvent polarity also for retinyl palmitate, retinaldehyde, lutein, and zeaxanthin. The local concentrations of carotenoids and vitamin E are likely to vary in different intracellular compartments. Alpha-tocopherol can be expected to be present in lipid membranes, but it is not so clear whether this is the case for lutein and zeaxanthin, which can bind to proteins such as pi isoform of glutathione S-transferase (GSTP1) or steroidogenic acute regulatory domain 3 (StARD3) [39]. Binding of carotenoid to protein may greatly influence their diffusion and antioxidant properties. While substantial progress has been made in understanding the selective uptake and accumulation of lutein and zeaxanthin in the retina [76,77,78,79,80], it is still unclear if free lutein and/or zeaxanthin are present in the POS or RPE or if they are bound to proteins.

The results of this study are of particular importance for diseases associated with delayed clearance of retinaldehydes and/or increased oxidative stress in the retina. It can be expected that the delayed clearance of retinaldehydes occurs in Stargardt’s disease caused by mutations in ABCA4 or RDH8 [25,26,27]. Upon absorption of ultraviolet or blue light, 11-*cis*- and all-*trans*-retinaldehydes can form an excited triplet state, which in the presence of oxygen can transfer the excess of energy to oxygen, generating an excited form of oxygen, singlet oxygen, or can transfer an electron generating superoxide anion radical [9,10,65]. Oxidative degradation of all-*trans*-retinaldehyde leads to a mixture of products retaining the ability to generate singlet oxygen and exhibiting increased (photo)toxic properties to cultured RPE cells [24]. These reactive oxygen species can induce oxidative damage to lipids, nucleic acids, and proteins. The latter includes ABCA4, which, as a result of exposure to visible light in the presence of all-*trans*-retinaldehyde, can be inactivated as an enzyme and form aggregates [81]. While it is not clear whether the damage to ABCA4 is due to reactive oxygen species produced by photoexcited retinaldehyde in the presence of oxygen or its oxidation products [9,10,24,65], it cannot be excluded that retinaldehyde oxidation products could be at least partly involved.

Mouse models of Stargardt’s disease: single gene knockouts: *abca4-/-* and particularly *rdh8-/-*, or double knockout *abca4(-/-)rdh8(-/-)* mice exhibit delayed clearance of all-*trans*-retinaldehyde after exposure to light and increased susceptibility to retinal degeneration manifesting itself as loss of photoreceptive neurons, which is accelerated when animals are raised in cyclic light as opposed to when rearing in the dark [2,64,82,83,84,85,86,87,88]. There is a growing body of evidence demonstrating that these single-knockout and double-knockout mice exhibit high susceptibility to retinal injury by acute exposure to light [64,89,90,91,92,93,94,95,96,97,98,99,100,101,102,103,104,105,106,107,108,109,110,111]. Also, the loss of function of retinol dehydrogenase 12 (RDH12), which is expressed in photoreceptor inner segments where it reduces retinaldehydes to retinol, leads to the increased susceptibility of the retina to light-induced injury [112]. Triple knockout mice *abca4(-/-)rdh8(-/-)rdh12(-/-)* raised under cyclic light develop retinal degeneration at the age of 6 weeks, whereas at least 3 months are needed to demonstrate a similar level of retinal degeneration for double knockout mice *abca4(-/-)rdh8(-/-)* [113]. Mutations of *RDH12* can cause early onset severe cone-rod dystrophy, Leber congenital amaurosis, retinitis pigmentosa, and pseudocoloboma-like maculopathy [114,115,116,117,118].

As mentioned in the Introduction, the AMD retina is at increased risk of oxidative damage due to an increased content of iron, and several products of lipid and protein oxidation have been detected there [11,12,13]. It can be expected that, in in the aged retina, and even more so in AMD, the retinoid trafficking and rapid reduction of all-*trans*-retinaldehyde to all-*trans*-retinol can be impaired due to the disordered structure of photoreceptor outer segments [66]. This is consistent with delayed dark adaptation observed in elderly patients and even more so in AMD. One of the potential factors responsible for the delayed dark adaptation could be an inadequate supply of 11-*cis*-retinaldehyde to visual pigments. This hypothesis of decreased availability of 11-*cis*-retinaldehyde was investigated by Hanneken et al. who found no difference in the ability of normal and AMD eyes to recover all rhodopsin during dark-adaptation post-mortem, suggesting that availability of 11-*cis*-retinaldehyde is not a limiting factor [119]. However, they allowed several hours for the dark adaptation to occur, so delayed trafficking of 11-*cis*-retinaldehydes to the convoluted POS in AMD retina cannot be excluded. Also, such convoluted POS may impair the trafficking of enzymatic co-factors, namely ATP and NADPH, needed by ABCA4 and RDH8 for the flipping of all-*trans*-retinaldehyde to the outer leaflet of the disk lipid membrane and its reduction to all-*trans*-retinol. Then all-trans-retinol needs to diffuse outside the convoluted POS to bind to a chaperone protein, interphotoreceptor retinoid binding protein (IRBP), for its transport to the RPE. The delayed clearance of retinoids from POS in the environment exposed to light and rich in polyunsaturated fatty acids and iron may allow for their oxidation [11,67]; therefore, adequate antioxidant protection is essential to prevent the downstream deleterious effects.

The results of this study demonstrating the effectiveness of lutein, zeaxanthin, α-tocopherol, and ascorbate in scavenging retinoid cation radicals are consistent with the results of a large multicentre randomized placebo-controlled clinical trial Age-Related Eye Disease Study 2 (AREDS2) showing that a daily supplementation with zinc, vitamins C and E, lutein, and zeaxanthin slows down the progression of AMD, particularly in persons with low dietary intake of these antioxidants [31,32,33]. AREDS2 also demonstrated that including lutein and zeaxanthin in the antioxidant mixture brings about a greater benefit than having no carotenoids or β-carotene. It is thought that lutein, zeaxanthin, α-tocopherol, and ascorbate can exert their beneficial effect by scavenging lipid-derived free radicals and superoxide and, in the case of lutein and zeaxanthin, by quenching of singlet oxygen and excited states of photosensitizers as well as by absorption of blue light in the innermost layers of the retina in the centre of the macula, thereby preventing blue light reaching POS and RPE in this area [38,39,68,69,120]. The results of this study demonstrate that the scavenging of retinoid cation radicals can be added to the repertoire of antioxidant actions of AREDS2 antioxidants.

## 4. Materials and Methods

### 4.1. Materials

Lutein and zeaxanthin were from DSM Nutritional Products AG (Basel, Switzerland). Retinoids (all-*trans*-retinaldehyde, all-*trans*-retinol, all-*trans*-retinyl palmitate), α-tocopherol, ascorbic acid, taurine, dihydroxyphenylalanine (dopa), Triton X-100, potassium bromide, disodium hydrogen phosphate, potassium dihydrogen phosphate, and thiocyanate were purchased from Sigma Aldrich (St. Louis, MO, USA). Benzene was purchased from VWR BDH Chemicals (Reading, UK). Cylinders with gases: nitrous oxide, argon, and nitrogen were purchased from BOC Ltd. (Woking, UK).

### 4.2. Synthesis of Dopa-Melanin

Dopa-melanin was used as a model of eumelanin present in the RPE [43,76]. It was synthesized by autooxidation of dopa as described previously [71,121]. For calculations of bimolecular rate constants of scavenging, the molecular weight of dopa-melanin monomer of 150 g/mol was used.

### 4.3. Preparation of Triton X-100 Micelles with Retinoids

All handling of retinoids was done under dim light. The retinoids and Triton X-100 were solubilized in argon-saturated benzene in a round flask, and the solvent was removed on rotary evaporator (Buchi, Flawil, Switzerland) while a thin film of retinoids and Triton X-100 was formed on the inner surface of the flask. To saturate the benzene with argon, a round flask was attached to the rotary evaporated, the pressure was lowered to 40 mbar, and then argon was allowed to the system to reach the atmospheric pressure. The remnants of benzene were removed from the flask under vacuum. Before removing the flask from the rotary evaporator, argon was allowed to fill the flask until the atmospheric pressure was reached. Then the flask was detached from the rotary evaporator and closed with a stopper. Just before the experiments, the solution of 0.1 M KBr in 10 mM phosphate buffer was slowly added to allow for formation of Triton-X micelles with incorporated retinoids.

### 4.4. Generation of Retinoid Cation Radicals and Monitoring Their Interaction with Antioxidants

The retinoid radical cations were generated by pulse radiolysis of nitrous oxide-saturated benzene with 1 mM retinoids or of aqueous buffered solution of potassium bromide where 1 mM retinoids were solubilized in 2% Triton X-100 micelles as described previously [18,28]. To saturate samples with nitrous oxide, a solution was placed in 25, 50, or 100 mL flasks with caps allowing for the flow of connected gas to the bottom of the flask and gently bubbling for 30 min to replace the air. The taps on the cap were closed and the flask was transferred to the pulse radiolysis room where the taps were connected to the optical cell and nitrogen gas. For radiolysis of the solution, a 20 to 500 ns pulse of electrons from a 9–12 MeV linear particle accelerator (Metropolitan-Vickers, Manchester, UK) or a 12-MeV RDL, 3-GHz electron linear accelerator with a pulse duration from 0.22 to 2 µs was used. As targeted a quartz flow-through cell with an optical path-length of 2.5 cm filled with a solution in benzene or water was used. The solution was supplied to the cell under pressure of nitrogen from a stock flask. Thiocyanate was used as a dosimeter. Changes in the transmittance of the solution at a selected wavelength were monitored at 90° by transient absorption spectroscopy.

In the case of benzene, the pulse of electrons caused the ionization of the solvent and the formation of benzene cation radicals, free electrons, and excited states. The electrons were scavenged by nitrous oxide, resulting in the formation of dinitrogen and anion radical of atomic oxygen. The oxygen anion radical was scavenged by benzene forming a radical anion adduct. The radical anion adduct was scavenged also by nitrous oxide, and the product reacted with benzene forming stable products. The cation radicals of benzene abstracted electrons from retinoids forming retinoid cation radicals [18,28].

In the case of the aqueous solution of 0.1 M potassium bromide and 10 mM phosphate buffer, pH7.0, radiolysis of water by a pulse of electrons generated hydroxyl radicals and solvated electrons. Scavenging of solvated electrons by nitrous oxide in the presence of water generated more hydroxyl radicals. Hydroxyl radicals were scavenged by bromide anions generating bromine radicals. The interaction of bromide anions with bromine radicals resulted in the formation of a highly oxidizing dibromine radical anion. The interaction of the dibromine radical anion with retinoids solubilized in Triton X-100 micelles led to the formation of retinoid cation radicals [18].

To study the interaction between retinoid cation radicals with antioxidants, the selected antioxidant was present in the solution at concentrations up to 0.1 mM, and their effect on the kinetics of retinoid cation radicals was determined. In addition, for experiments involving lutein or zeaxanthin, we also monitored the formation of the carotenoid cation radical.

## 5. Conclusions

In conclusion, we have determined the bimolecular rate constant of scavenging cation radicals of retinoids involved in vision by lutein, zeaxanthin, dopa-melanin, and, in case of retinyl palmitate cation radicals, also by ascorbate and α-tocopherol. No increase in the rate of decay of retinoid cation radicals was observed in the presence of taurine, so it has been evaluated that the bimolecular rate constants of scavenging cation radicals of retinoids by taurine must be smaller than 2 × 10^7^ M^−1^s^−1^. Lutein scavenges cation radicals of all three retinoids with the bimolecular rate constants approaching the diffusion-controlled limits, while zeaxanthin is only 1.4–1.6-fold less effective. Despite lutein exhibiting greater scavenging rate constants of retinoid cation radicals than other antioxidants, the greater concentrations of ascorbate in the retina suggest that vitamin C may be the main protectant of the visual cycle retinoids from oxidative degradation. Vitamin E may play a substantial role in the protection of retinaldehyde but is relatively inefficient in the protection of retinol or retinyl palmitate. While the protection of retinoids by lutein and zeaxanthin appears inefficient in the retinal periphery, it can be quite substantial in the macula. Although the determined rate constants of the interaction of the cation radicals of retinol and retinaldehyde with synthetic dopa-melanin are relatively low, the high concentration of melanin in the RPE melanosomes suggest that melanin can be effective in scavenging the retinoid cation radicals in proximity to melanin-containing pigment granules.

## Figures and Tables

**Figure 1 ijms-25-00506-f001:**
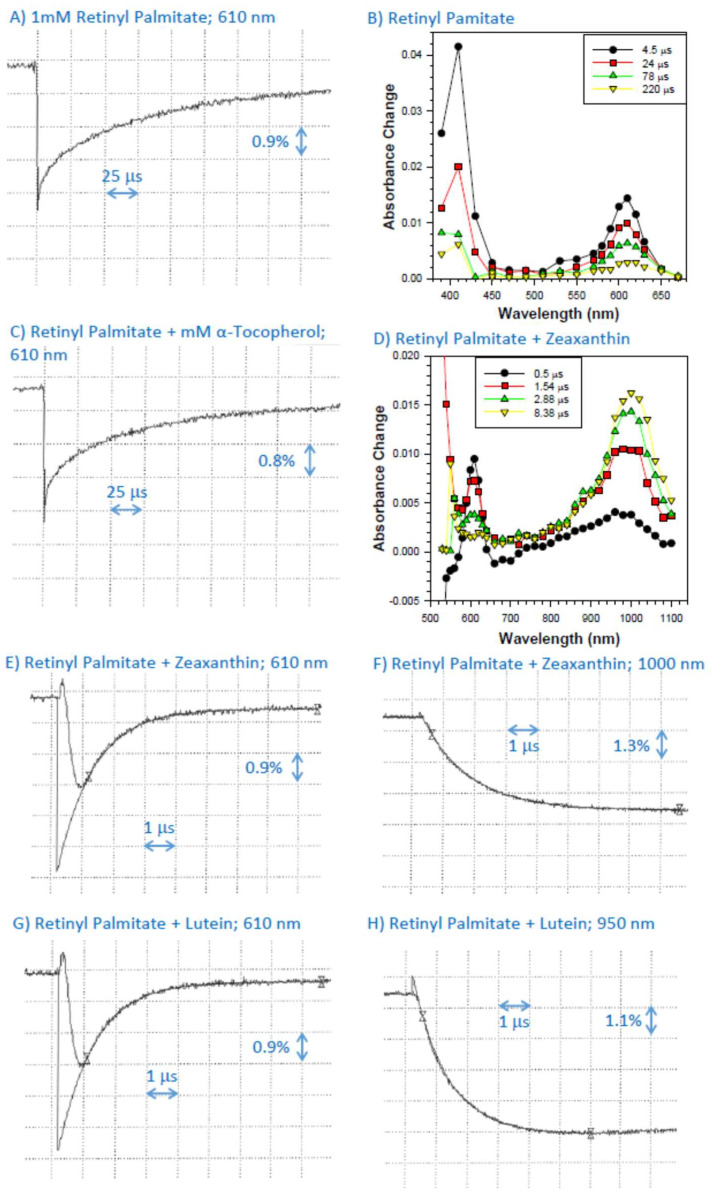
Representative kinetics of the formation and decay of the transient species monitored at 610 nm after pulse radiolysis of N_2_O-saturated benzene with solubilized 1 mM of retinyl palmitate (**A**) and transient absorption spectra at indicated times after the pulse radiolysis of that solution (**B**). (**C**) Representative kinetics of the formation and decay of the transient species monitored at 610 nm after pulse radiolysis of N_2_O-saturated benzene with solubilized 1 mM of retinyl palmitate in the presence of 0.1 mM α-tocopherol. (**D**) Transient absorption spectra at indicated times after the pulse radiolysis of N_2_O-saturated benzene with solubilized 1 mM of retinyl palmitate and 0.1 mM zeaxanthin, and representative kinetics of the formation and decay of transient species monitored at 610 nm (**E**) and 1000 nm (**F**) after pulse radiolysis of that solution. (**G**,**H**) Representative kinetics of the formation and decay of transient species monitored at 610 nm (**G**) and 950 nm (**H**) after pulse radiolysis of N_2_O-saturated benzene with solubilized 1 mM of retinyl palmitate in the presence of 0.1 mM lutein.

**Figure 2 ijms-25-00506-f002:**
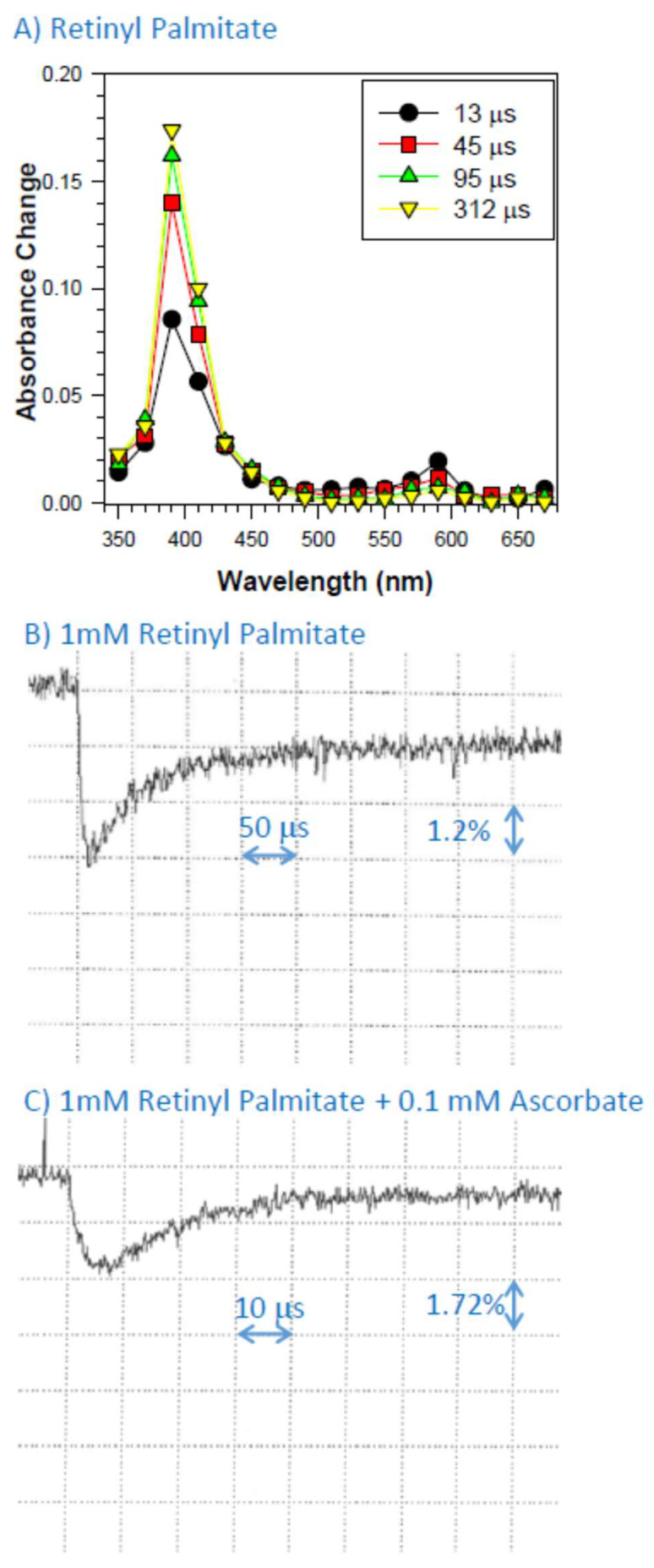
Transient absorption spectra after pulse radiolysis of aqueous solution saturated with N_2_O and containing 10 mM phosphate, pH 7, 0.1 M KBr, and 1 mM retinyl palmitate incorporated in 2% Triton X-100 micelles (**A**) and a representative kinetics of the formation and decay of retinyl palmitate cation radicals monitored at 590 nm after pulse radiolysis of that solution in the absence (**B**) and presence of 0.1 mM of ascorbate (**C**).

**Figure 3 ijms-25-00506-f003:**
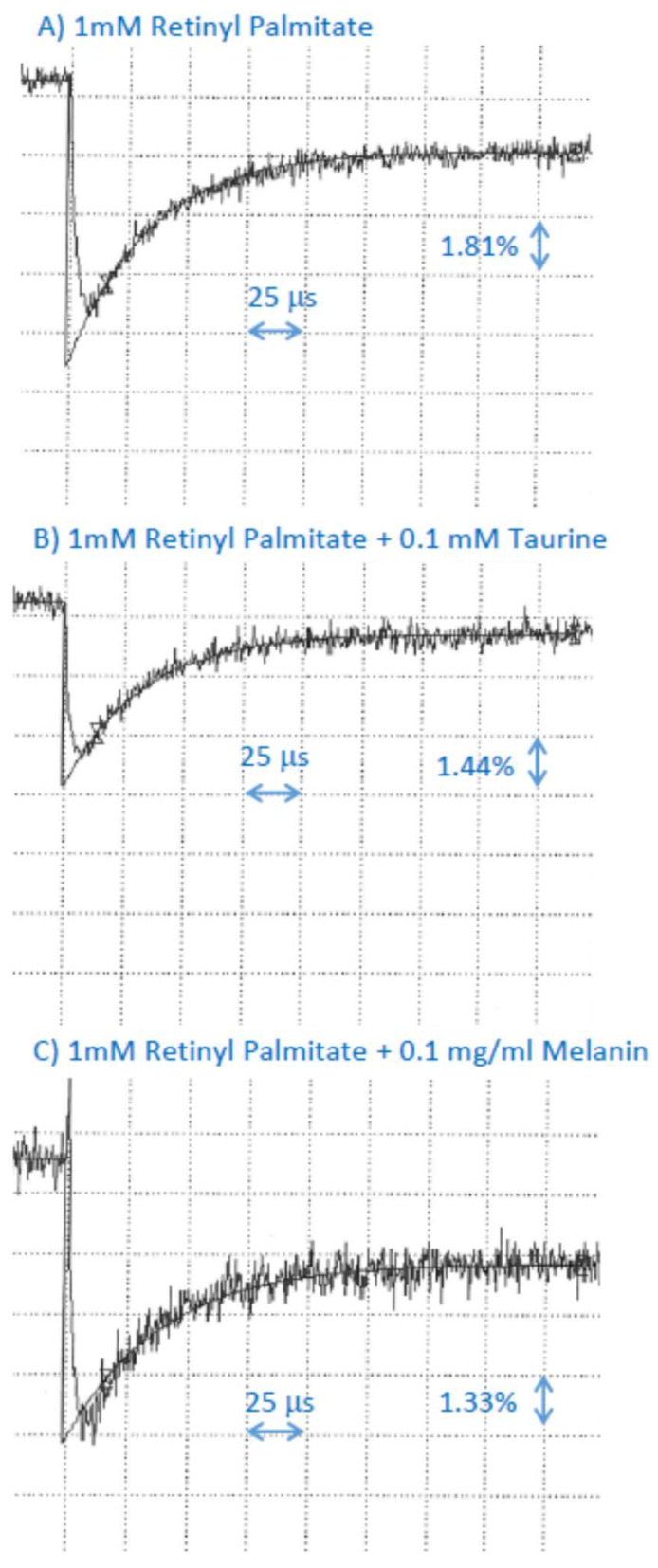
Representative kinetics of the formation and decay of retinyl palmitate cation radicals monitored at 590 nm after pulse radiolysis of aqueous solution saturated with N_2_O and containing 10 mM phosphate, pH 7, 0.1 M KBr, and 1 mM retinyl palmitate incorporated in 2% Triton X-100 micelles, in the absence (**A**) and presence of 0.1 mM of taurine (**B**) or 0.1 mg/mL (equivalent to 0.67 mM monomers) dopa-melanin (**C**).

**Figure 4 ijms-25-00506-f004:**
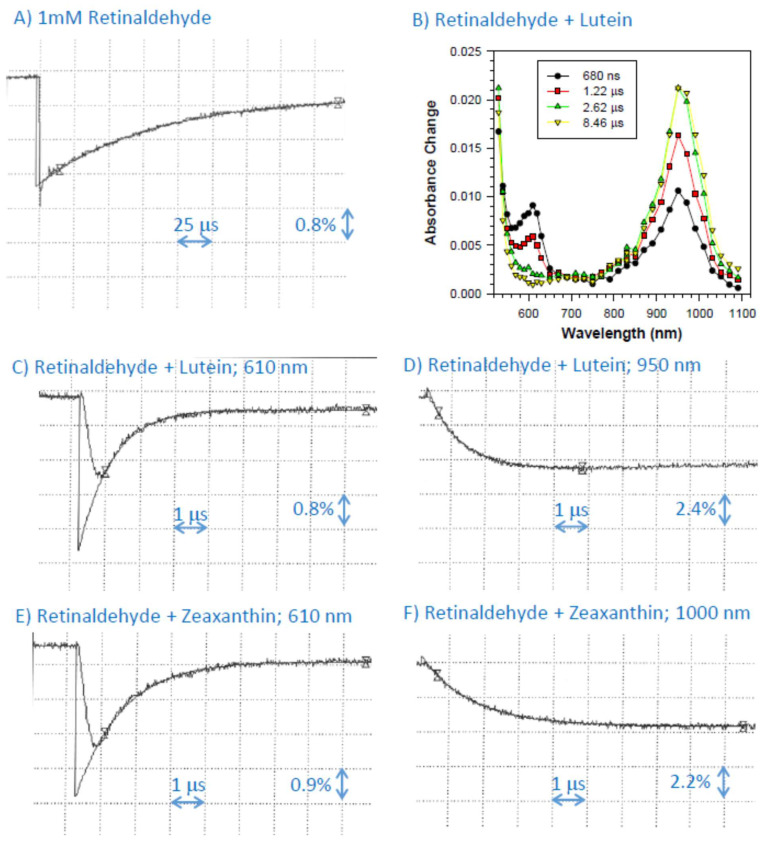
(**A**) Representative kinetics of the formation and decay of the transient species monitored at 610 nm after pulse radiolysis of N_2_O-saturated benzene with solubilized 1 mM of retinaldehyde. (**B**–**D**): Transient absorption spectra at indicated times after the pulse radiolysis of N_2_O-saturated benzene with solubilized 1 mM of retinaldehyde and 0.1 mM lutein (**B**) and representative kinetics of the of the formation and decay of the transient species after the pulse radiolysis of that solution monitored at 610 nm (**C**) and 950 nm (**D**) after pulse radiolysis of that solution. (**E**,**F**) Representative kinetics of the formation and decay of transient species monitored at 610 nm (**E**) and 1000 nm (**F**) after pulse radiolysis of N_2_O-saturated benzene with solubilized 1 mM of retinaldehyde in the presence of 0.1 mM zeaxanthin.

**Figure 5 ijms-25-00506-f005:**
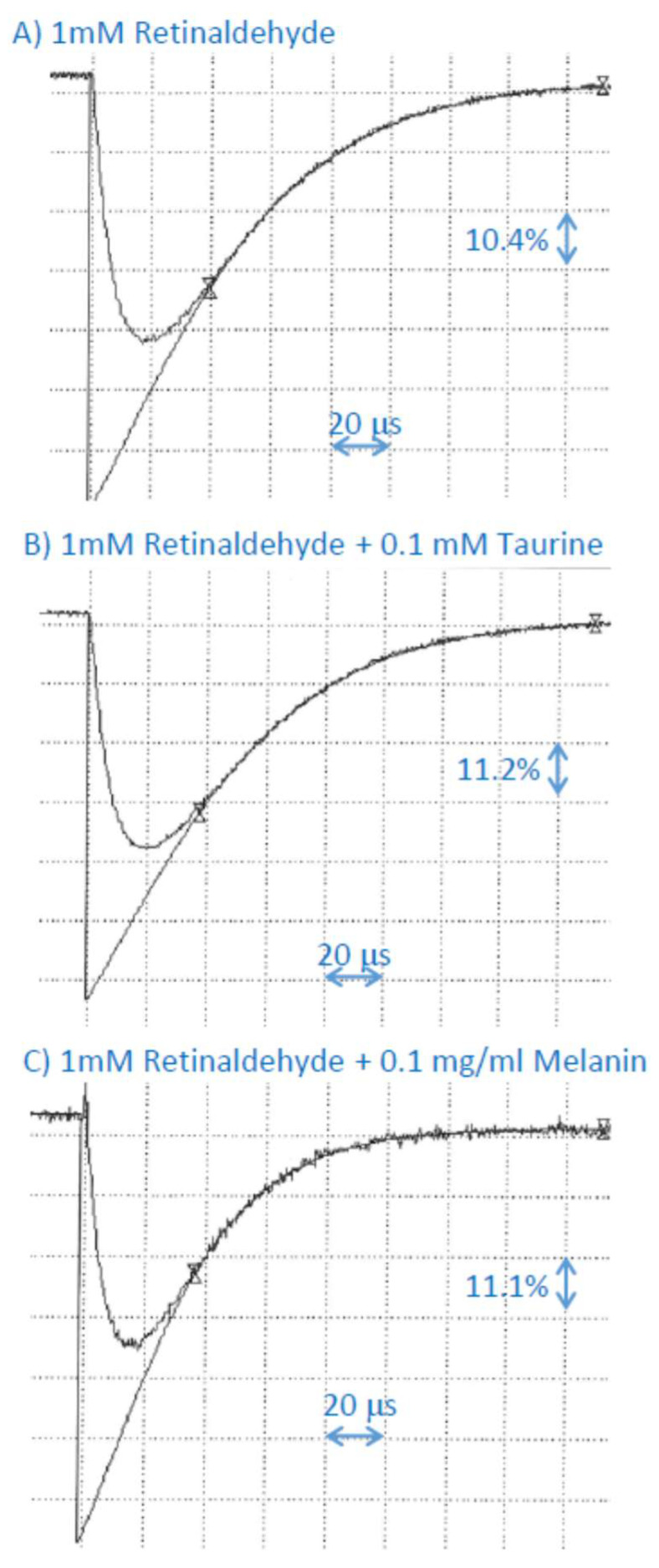
Representative kinetics of the formation and decay of retinaldehyde cation radicals monitored at 590 nm after pulse radiolysis of aqueous solution saturated with N_2_O and containing 10 mM phosphate, pH 7, 0.1 M KBr, and 1 mM retinaldehyde incorporated in 2% Triton X-100 micelles, in the absence (**A**) and presence of 0.1 mM of taurine (**B**) or 0.1 mg/mL (equivalent to 0.67 mM monomers) dopa-melanin (**C**).

**Figure 6 ijms-25-00506-f006:**
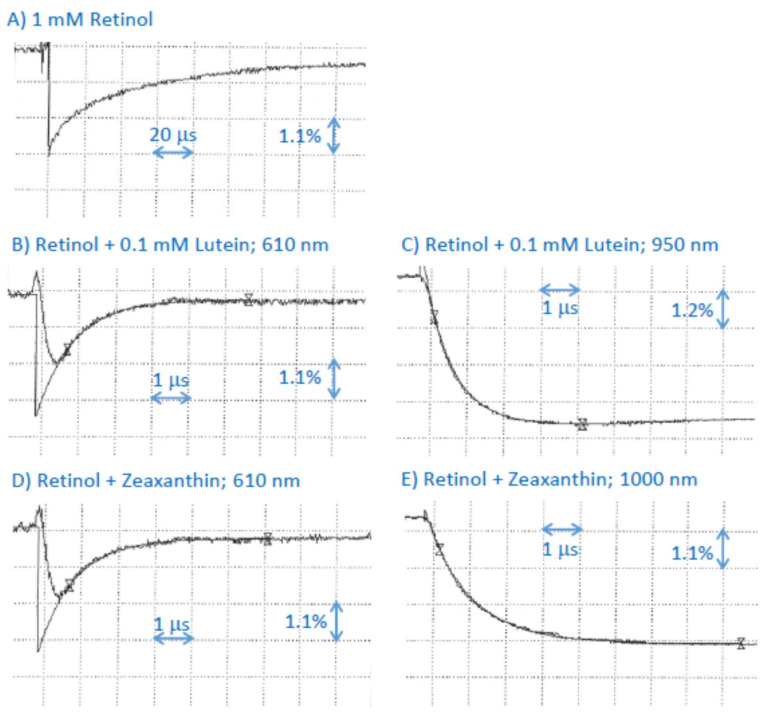
(**A**) Representative kinetics of the formation and decay of the transient species monitored at 610 nm after pulse radiolysis of N_2_O-saturated benzene with solubilized 1 mM of retinol. (**B**,**C**) Representative kinetics of the formation and decay of transient species monitored at 610 nm (**B**) and 950 nm (**C**) after pulse radiolysis of N_2_O-saturated benzene with solubilized 1 mM of retinol in the presence of 0.1 mM lutein. (**D**,**E**) Representative kinetics of the formation and decay of transient species monitored at 610 nm (**D**) and 1000 nm (**E**) after pulse radiolysis of N_2_O-saturated benzene with solubilized 1 mM of retinaldehyde in the presence of 0.1 mM zeaxanthin.

**Figure 7 ijms-25-00506-f007:**
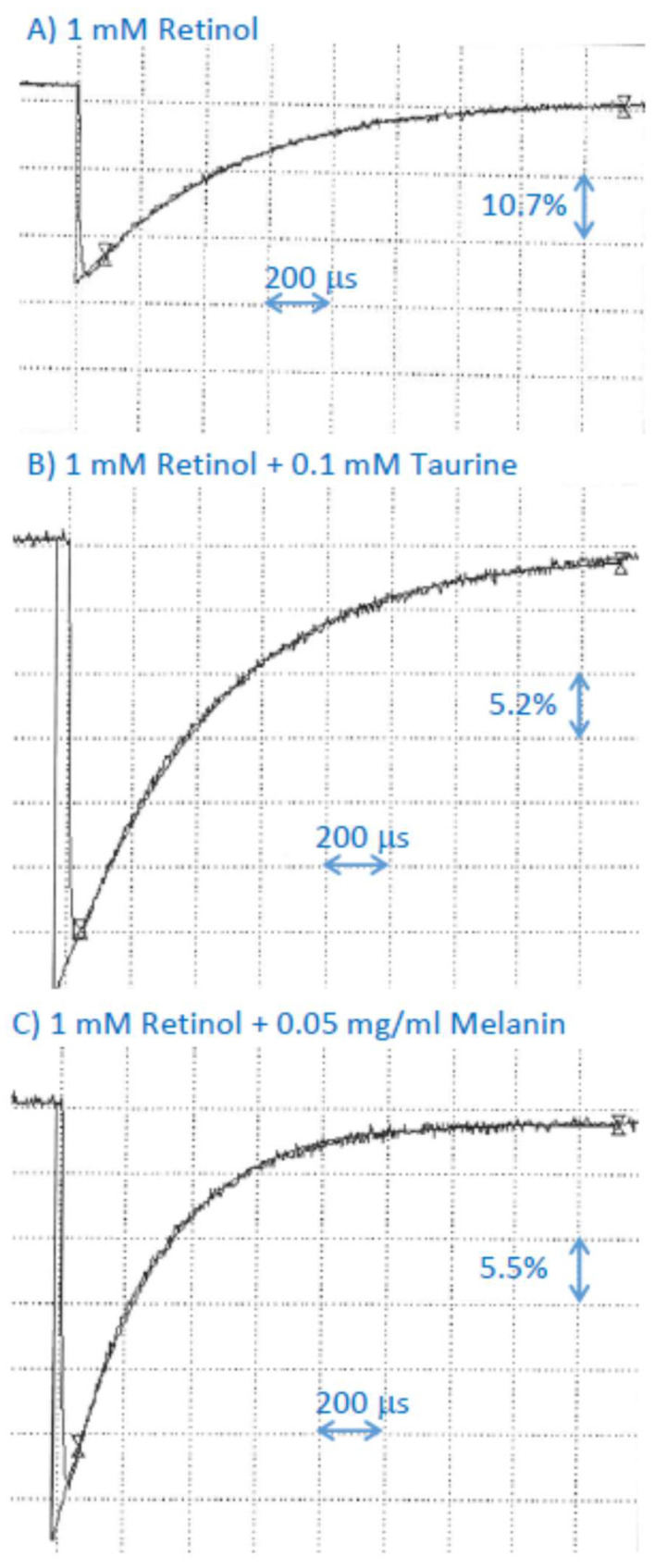
Representative kinetics of the formation and decay of retinol cation radicals monitored at 590 nm after pulse radiolysis of aqueous solution saturated with N_2_O and containing 10 mM phosphate, pH 7, 0.1 M KBr, and 1 mM retinol incorporated in 2% Triton X-100 micelles, in the absence (**A**) and presence of 0.1 mM of taurine (**B**) or 0.1 mg/mL (equivalent to 0.67 mM monomers) dopa-melanin (**C**).

**Table 1 ijms-25-00506-t001:** Bimolecular rate constants of scavenging of retinoid cation radicals by lutein, zeaxanthin, vitamin E (α-tocopherol), vitamin C (ascorbic acid), taurine, and melanin (dopa-melanin). To determine the bimolecular rate constants of interactions between lipophilic antioxidants (lutein, zeaxanthin, and α-tocopherol) and retinoid cation radicals, both reactants were solubilized in benzene. To determine the bimolecular rate constants of interactions between retinoid cation radicals and hydrophilic antioxidants (ascorbate, taurine, dopa-melanin), the retinoids were incorporated into Triton X-100 micelles, whereas the hydrophilic antioxidants were solubilized directly in 10 mM of phosphate buffer pH 7.0. ^a^ from [21]; ^b^ from [28]; ^c^ from [18]; ^d^ from [22].

	Bimolecular Rates of Scavenging of Retinoid Radical Cations (10^9^ M^−1^s^−1^)		
	Retinyl Palmitate	Retinaldehyde	Retinol
Lutein	8.85 ± 0.25	11.5 ± 1.4	12.6 ± 0.4
Zeaxanthin	6.39 ± 0.02	6.5 ± 0.3	7.9 ± 0.3 5.76 ± 0.50 in methanol ^a^ 2.50 ± 0.20 in benzonitrile ^a^
α-Tocopherol	0.027 ± 0.003	8.0 ± 0.3 ^b^	0.080 ± 0.045 ^a^
Ascorbate	0.58 ± 0.10	0.73 ^c^ 0.53 ± 0.08 in methanol ^d^	0.12 ^c^
Taurine	<0.02	<0.01	<0.002
Dopa-melanin	<0.02	0.016 ± 0.008	0.0051 ± 0.0001

**Table 2 ijms-25-00506-t002:** Effectiveness of scavenging of retinyl palmitate cation radicals in RPE, which is the main storage site of retinyl esters in the retina. The multiplication products are based on average concentrations of lutein and zeaxanthin of the macular and peripheral RPE isolated from retinas of people with normal lutein/zeaxanthin intake, increased intake, and the highest reported and on average and maximal concentrations of vitamin E in the RPE. It is not clear what the concentrations of ascorbate are in the RPE; therefore, the calculations are done for 1 mM detected in brain glial cells, 2 mM in the vitreous body, and 10 mM in the brain neurons. See the main body of text for more details on concentrations of antioxidants in the retina.

	Effectiveness of Scavenging of Retinyl Palmitate Radical Cations in the RPE Scavenging Rate Constant × Antioxidant Concentration (10^3^ s^−1^)			
	Lutein/Zeaxanthin			
	Normal Intake	Increased Intake		Maximum
[Macular RPE lutein] (µM)	33.8	97.9		172
[Macular RPE lutein]k_Q_	299	866		1520
[Macular RPE zeaxanthin] (µM)	22.5	65.3		133
[Macular RPE zeaxanthin]k_Q_	144	417		848
[Peripheral RPE lutein] (µM)	1.14	3.30		5.8
[Peripheral RPE lutein]k_Q_	10.1	29.2		51.3
[Peripheral RPE zeaxanthin] (µM)	0.32	0.92		1.87
[Peripheral RPE zeaxanthin]k_Q_	2.0	5.9		12.0
	Vitamin E (α-tocopherol)			
	Average		Maximum	
[RPE Vitamin E] (µM)	115		230	
[RPE Vitamin E]k_Q_	3.11		6.21	
	Vitamin C (ascorbate)			
[Ascorbate] (mM)	1 mM	2 mM		10 mM
[Ascorbate]k_Q_	580	1160		5800

**Table 3 ijms-25-00506-t003:** Effectiveness of scavenging of retinol cation radicals by antioxidants in pathways of retinol trafficking in the retina. The multiplication products are based on average concentrations of lutein and zeaxanthin of the macular and peripheral RPE and POS from retinas of people with normal lutein/zeaxanthin intake, increased intake, and the highest reported on average and maximal concentrations of vitamin E in the RPE and neural retinas dissected from macula and periphery. It is not clear what the concentrations of ascorbate are in the RPE; therefore, the calculations are done based on ascorbate concentrations of 1 mM in brain glial cells, 2 mM concentration of ascorbate in the vitreous body, and 10 mM concentration of ascorbate in the brain neurons. See the main body of text for more details on concentrations of antioxidants in the retina.

	Effectiveness of Scavenging of Retinol Radical Cations in the RPE and Neural Retina Scavenging Rate Constant × Antioxidant Concentration (10^3^ s^−1^)			
	Lutein/Zeaxanthin			
	Normal Intake	Increased Intake		Maximum
[Macular RPE lutein] (µM)	33.8	97.9		172
[Macular RPE lutein]k_Q_	425	1233		2169
[Macular RPE zeaxanthin] (µM)	22.5	65.3		133
[Macular RPE zeaxanthin]k_Q_	178	515		1049
[Peripheral RPE lutein] (µM)	1.14	3.30		5.8
[Peripheral RPE lutein]k_Q_	14	42		73
[Peripheral RPE zeaxanthin] (µM)	0.32	0.92		1.87
[Peripheral RPE zeaxanthin]k_Q_	2.5	7.3		14.8
[Macular POS lutein] (µM)	18.8	54.5		95.9
[Macular POS lutein]k_Q_	237	687		1209
[Macular POS zeaxanthin] (µM)	13.7	39.7		80.7
[Macular POS zeaxanthin]k_Q_	108	314		638
[Peripheral POS lutein] (µM)	0.63	1.84		3.23
[Peripheral POS lutein]k_Q_	8.0	23.1		40.7
[Peripheral POS zeaxanthin] (µM)	0.19	0.56		1.14
[Peripheral POS zeaxanthin]k_Q_	1.5	4.4		9.0
	Vitamin E (α-tocopherol)			
	Average		Maximum	
[RPE Vitamin E] (µM)	115		230	
[RPE Vitamin E]k_Q_	9.2		18.4	
[neural vitamin E; macula] (µM)	46		78	
[neural vitamin E; macula]k_Q_	3.7		6.3	
[neural vitamin E; periphery] (µM)	77		124	
[neural vitamin E; periphery]k_Q_	6.1		9.9	
	Vitamin C (ascorbate)			
[Ascorbate] (mM)	1 mM	2 mM		10 mM
[Ascorbate]k_Q_	120	240		1200
	Melanin in RPE melanosomes			
[Young RPE melanin] (mM)	398			
[Young RPE melanin]k_Q_	2030			
[Old RPE melanin] (mM)	310			
[Old RPE melanin]k_Q_	1581			

**Table 4 ijms-25-00506-t004:** Effectiveness of scavenging of retinaldehyde cation radicals in pathways of retinal trafficking in the retina. See Table 3 caption for further details.

	Effectiveness of Scavenging of Retinaldehyde Radical Cations in the RPE and Neural Retina Scavenging Rate Constant × Antioxidant Concentration (10^3^ s^−1^)			
	Lutein/Zeaxanthin			
	Normal Intake	Increased Intake		Maximum
[Macular RPE lutein] (µM)	33.8	97.9		172
[Macular RPE lutein]k_Q_	388	1126		1979
[Macular RPE zeaxanthin] (µM)	22.5	65.3		133
[Macular RPE zeaxanthin]k_Q_	146	424		863
[Peripheral RPE lutein] (µM)	1.14	3.30		5.8
[Peripheral RPE lutein]k_Q_	13	38		67
[Peripheral RPE zeaxanthin] (µM)	0.32	0.92		1.87
[Peripheral RPE zeaxanthin]k_Q_	2.1	6.0		12.2
[Macular POS lutein] (µM)	18.8	54.5		95.9
[Macular POS lutein]k_Q_	217	627		1103
[Macular POS zeaxanthin] (µM)	13.7	39.7		80.7
[Macular POS zeaxanthin]k_Q_	89	258		525
[Peripheral POS lutein] (µM)	0.63	1.84		3.23
[Peripheral POS lutein]k_Q_	7.3	21.1		37.1
[Peripheral POS zeaxanthin] (µM)	0.19	0.56		1.14
[Peripheral POS zeaxanthin]k_Q_	1.3	3.6		7.4
	Vitamin E (α-tocopherol)			
	Average		Maximum	
[RPE Vitamin E] (µM)	115		230	
[RPE Vitamin E]k_Q_	920		1840	
[neural vitamin E; macula] (µM)	46		78	
[neural vitamin E; macula]k_Q_	367		627	
[neural vitamin E; periphery] (µM)	77		124	
[neural vitamin E; periphery]k_Q_	613		991	
	Vitamin C (ascorbate)			
[Ascorbate] (mM)	1 mM	2 mM		10 mM
[Ascorbate]k_Q_	730	1460		7300
	Melanin in RPE melanosomes			
[Young RPE melanin] (mM)	398			
[Young RPE melanin]k_Q_	6368			
[Old RPE melanin] (mM)	310			
[Old RPE melanin]k_Q_	4960			

## Data Availability

All data are presented in the manuscript.

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
