# Peer review of "Scavenging of Cation Radicals of the Visual Cycle Retinoids by Lutein, Zeaxanthin, Taurine, and Melanin"

_ijms, 2023, doi:10.3390/ijms25010506_

Round 1

Reviewer 1 Report

Comments and Suggestions for Authors

Rozanowska et al, shows the role of Lutein, Zeaxanthin, Taurine and Melanin in scavenging cation radicals of the visual cycle retinoids. The retinal retinoids are under constant threat of oxidation. In this study the authors evaluate the biomolecular rate constant of lutein, zeaxanthin, dopa-melanin, ascorbate and alpha tocopherol. Lutein being the higher concentration in the visual cycle of retinoid might play a role in the protection against oxidation. The role of lutein and zeaxanthin might be different depending on the region of the eye. Vitamin C due to its concentration in the retina might be the protectant of the visual cycle of retinoids from oxidation.

The study is interesting but few minor suggestions and questions for acceptance.

1.     Conclusion that ascorbate might be the protectant of visual cycle of retinoids just based on the concentration. The evidence provided shows that other molecules Lutein, Zeaxanthin are more potent than ascorbate in scavenging radical cations. How do you explain this caveat? Perhaps time as variable in describing potency inputting the concentration of the antioxidants in real time would be an alternative.

2.     Please include the concentration of the antioxidants in the RPE region, macula, retina for easier comparison in a table format. Also the table needs to be clearer for easier understanding. Perhaps split the table with scavenging rate constant vs antioxidant concentration. The current format is confusing to read.

Comments on the Quality of English Language

The paper is understandable and its easier to read but needs few minor revisions. 

Author Response

We thank the Reviewer 1 for reading the manuscript and their comments.

Rozanowska et al, shows the role of Lutein, Zeaxanthin, Taurine and Melanin in scavenging cation radicals of the visual cycle retinoids. The retinal retinoids are under constant threat of oxidation. In this study the authors evaluate the biomolecular rate constant of lutein, zeaxanthin, dopa-melanin, ascorbate and alpha tocopherol. Lutein being the higher concentration in the visual cycle of retinoid might play a role in the protection against oxidation. The role of lutein and zeaxanthin might be different depending on the region of the eye. Vitamin C due to its concentration in the retina might be the protectant of the visual cycle of retinoids from oxidation.

The study is interesting but few minor suggestions and questions for acceptance.

  1. Conclusion that ascorbate might be the protectant of visual cycle of retinoids just based on the concentration. The evidence provided shows that other molecules Lutein, Zeaxanthin are more potent than ascorbate in scavenging radical cations. How do you explain this caveat? Perhaps time as variable in describing potency inputting the concentration of the antioxidants in real time would be an alternative.

The explanation is provided in Table 2 where the products of concentrations of antioxidants and the quenching rate constants are included. These products are the best way to compare the antioxidant efficacy of different antioxidants in different retinal areas.

  1. Please include the concentration of the antioxidants in the RPE region, macula, retina for easier comparison in a table format. Also the table needs to be clearer for easier understanding. Perhaps split the table with scavenging rate constant vs antioxidant concentration. The current format is confusing to read.

Table 2 includes concentrations of antioxidants in different retinal areas. We have highlighted in bright green all cells with products of multiplication of antioxidant concentrations and quenching rate constants.

Reviewer 2 Report

Comments and Suggestions for Authors

The authors in the article titled “Scavenging of Cation Radicals of the Visual Cycle Retinoids by Lutein, Zeaxanthin, Taurine, and Melanin‚‚ presented the results of determination of the bimolecular rate constant of scavenging cation radicals of retinoids involved in vision by lutein, zeaxanthin, dopa-melanin, and, in case of retinyl palmitate cation radicals also by ascorbate and α-tocopherol. I recommend the paper for publishing after major revisions.

One notable aspect of the manuscript is the thoroughness with which you delve into certain sections, providing a comprehensive understanding of the mechanism. However, I would like to offer a suggestion for improvement. In some parts of the manuscript, particularly in table and figure titles, the detailed explanations, while informative, might benefit from a more concise presentation to enhance readability.

Additionally, I observed that the methodology section is comparatively brief. Providing more details about methodology, including specific details, could enhance the clarity and replicability of the study. A more comprehensive methodology would contribute to the robustness of your research.

Line 4. The way of naming the authors is not the same.

Line 21. The passive tense is better for scientific papers.

Line 22. In Title the authors mention Taurine, but in there is no information about it in abstract part.

Line 23. The authors should decide whether to write alpha or α.

Line 91. The passive tense is better for scientific papers.

Line 100. The passive tense is better for scientific papers.

Line 123-143. This paragraph should be in the results.

Line 154. The title of Figure 1 is too long. The detailed description should be in the methodology part.

Lines 157 and 159. The authors decide whether to write letters in brackets or with colon.

Line 173-179. This part should be in methodology part, not in table title. Statistical significance should be below the table. And must be understandable. Now it is not clear.

Table 1. Why the results are not the same for different samples? Zeaxanthin and ascorbate is in methanol, Zeaxanthin also in benzonitrile. What was the solvent for the other samples?

Line 206. W in while should be lowercase.

Line 253. The passive tense is better for this part.

Line 256. One space is extra before “In the presence…”.

Line 269. Description for figure 4 is too long.

Line 271: What is B-D:?

Line 307. Description for figure 6 is too long. And, the authors decide whether to write letters in brackets or with colon.

Line 357. The number 34 for references is missing. The authors mention the Sommerburg et al, but there are no references.

Line 362. One space is extra before “The highest…”

Line 380. Description for table 2 is too long. And the authors should not write “see the main body of text for more details”.

Line 461. One space is extra before “We also have…”.

Line 461. The passive tense is more suitable.

Line 465. Friedrichson et al is reference number 29, not 37.

Line 486. Description for table 3 is too long. And the authors should not write “see the main body of text for more details”.

Line 582. It is not appropriate to write “summary” in the discussion part. The authors should rewrite or move to conclusion.

Line 658. It does not matter to the scientific community that lutein and zeaxanthin were a gift.Line 660. Which “other chemicals”?

Line 665. The methodology part should be very precise. The authors should describe this process in more detail.

Line 670. Where was the Tritox X-100 obtained from?

Line 671. How was the argon-saturated benzene prepared.

Line 678. The full stop is not necessary after subtitles.

Line 680. How was the nitrous oxide-saturated benzene prepared?

Line 682. The authors should describe this process in more detail.

Line 684. Where was the thiocyanate obtained from?

Line 689. Where was the nitrous oxide obtained from?

Line 694. Where was the potassium bromide obtained from?

Line 709. In the conclusion part, again, taurine is not mention.

Comments on the Quality of English Language

English language is fine.

Author Response

We thank the Reviewer 2 for reading the manuscript and their comments.

The authors in the article titled “Scavenging of Cation Radicals of the Visual Cycle Retinoids by Lutein, Zeaxanthin, Taurine, and Melanin‚‚ presented the results of determination of the bimolecular rate constant of scavenging cation radicals of retinoids involved in vision by lutein, zeaxanthin, dopa-melanin, and, in case of retinyl palmitate cation radicals also by ascorbate and α-tocopherol. I recommend the paper for publishing after major revisions.

One notable aspect of the manuscript is the thoroughness with which you delve into certain sections, providing a comprehensive understanding of the mechanism. However, I would like to offer a suggestion for improvement. In some parts of the manuscript, particularly in table and figure titles, the detailed explanations, while informative, might benefit from a more concise presentation to enhance readability.

Additionally, I observed that the methodology section is comparatively brief. Providing more details about methodology, including specific details, could enhance the clarity and replicability of the study. A more comprehensive methodology would contribute to the robustness of your research.

 Line 4. The way of naming the authors is not the same.

We have moved the first initial of the last author to line 5.

Line 21. The passive tense is better for scientific papers.

As suggested, we replaced the phrase with “This study.”

Line 22. In Title the authors mention Taurine, but in there is no information about it in abstract part.

As suggested, the information about taurine has been added.

Line 23. The authors should decide whether to write alpha or α.

Two sentences were combined into one and “alpha” was replaced by α. We write “Alpha” only at the beginning of a sentence because capitalized α could be misread as “a.”

Line 91. The passive tense is better for scientific papers.

Changed to “It has been.”

Line 100. The passive tense is better for scientific papers.

Changed to “It has been.”

Line 123-143. This paragraph should be in the results.

The guidelines for manuscript preparation include this instruction for the Introduction: "Finally, briefly mention the main aim of the work and highlight the main conclusions.

Line 154. The title of Figure 1 is too long. The detailed description should be in the methodology part.

Fig. 1 is a multipanelled figure and there is no redundant information in the figure caption. 

Lines 157 and 159. The authors decide whether to write letters in brackets or with colon.

We have changed the formatting, so now it is consistent.

Line 173-179. This part should be in methodology part, not in table title. Statistical significance should be below the table. And must be understandable. Now it is not clear.

The solvents in which the scavenging rate constants from the literature were determined are given in the table. The scavenging rate constants reported in this paper were determined either in benzene (for lipophilic antioxidants) or in Triton-X solution in phosphate buffer (for hydrophilic antioxidants). It is important information and therefore it was included in the table caption.  

Table 1. Why the results are not the same for different samples? Zeaxanthin and ascorbate is in methanol, Zeaxanthin also in benzonitrile. What was the solvent for the other samples?

The diffusion rates are different in different solvents. The information about the solvent used in our experiments is provided in the table caption.

Line 206. W in while should be lowercase.

We have removed the redundant “This is.”

Line 253. The passive tense is better for this part.

We have replaced “our” with “the.”

Line 256. One space is extra before “In the presence…”.

Corrected as suggested.

Line 269. Description for figure 4 is too long.

Fig. 4 is a multipanelled figure and there is no redundant information in the figure caption. 

Line 271: What is B-D:?

Corrected E,F

Line 307. Description for figure 6 is too long. And, the authors decide whether to write letters in brackets or with colon.

Fig. 6 is a multipanelled figure and there is no redundant information in the figure caption. 

Line 357. The number 34 for references is missing. The authors mention the Sommerburg et al, but there are no references.

The reference number has been added.

Line 362. One space is extra before “The highest…”

The extra space has been removed.

Line 380. Description for table 2 is too long. And the authors should not write “see the main body of text for more details”.

All information in Tab. 2 caption is essential. We have changed the phrase so it reads “See the main body of text for more details on concentrations of antioxidants in the retina.”

Line 461. One space is extra before “We also have…”.

The extra space has been removed.

Line 461. The passive tense is more suitable.

Changed to “It has been.”

Line 465. Friedrichson et al is reference number 29, not 37.

We have included both references because reference 37 describes how, based on Friedrishson’s data of vitamin E contents, the concentrations of vitamin E were calculated.

Line 486. Description for table 3 is too long. And the authors should not write “see the main body of text for more details”.

All information in Table 3 caption is essential. We have changed the phrase so it is more specific “See the main body of text for more details on concentrations of antioxidants in the retina.”

Line 582. It is not appropriate to write “summary” in the discussion part. The authors should rewrite or move to conclusion.

We have provided a brief summary of the results in the discussion where it is more appropriate to have it than in the conclusion.

Line 658. It does not matter to the scientific community that lutein and zeaxanthin were a gift.Line 660. Which “other chemicals”?

We have removed “gift” from the Methods. All other chemicals used have been added.

Line 665. The methodology part should be very precise. The authors should describe this process in more detail.

The methods are based on methods used by us previously and described in detail in cited papers. We have added more details.

Line 670. Where was the Tritox X-100 obtained from?

The information has been added.

Line 671. How was the argon-saturated benzene prepared.

The information has been added.

Line 678. The full stop is not necessary after subtitles.

Removed as suggested.

Line 680. How was the nitrous oxide-saturated benzene prepared?

The information has been added.

Line 682. The authors should describe this process in more detail.

The information has been added.

Line 684. Where was the thiocyanate obtained from?

The information has been added.

Line 689. Where was the nitrous oxide obtained from?

The information has been added.

Line 694. Where was the potassium bromide obtained from?

The information has been added.

Line 709. In the conclusion part, again, taurine is not mention.

A sentence about taurine has been added to the conclusions.

Reviewer 3 Report

Comments and Suggestions for Authors

The authors have determined the bimolecular rate constant of scavenging cation radicals of retinyl palmitate, retinaldehyde and retinol by lutein, zeaxanthin, dopa-melanin, ascorbate and α-tocopherol.  The retinoid radical cations were generated by pulse radiolysis of saturated N2O in benzene or Triton X-100 micelles solubilized in phosphate buffer containing retinoids and cation radicals scavengers. Thiocyanate was used as an indicator and transmittance of solutions were measured by transient absorption spectroscopy. Lutein captured free radicals the fastest (8.85-12.6 x 109 M-1s-1) approaching the diffusion-limited limit, the next one was zeaxanthin (6.39-7.9 x 109 M-1s-1). The remaining results were very low, indicating a low rate of cation radical scavenging (<0.002-0.58 x 109 M-1s-1). Based on the conducted research, the authors try to explain the action of the investigated compounds in protection of retinoids involved in vision process. They concluded, among other things, that “While the protection of retinoids by lutein and zeaxanthin appears inefficient in the retinal periphery, it can be quite substantial in the macula”. The topic is original in terms of a new active compound and it address a specific gap in the field. The research and its results are interesting. The discussion of the results is good and refers to many publications. Studying the literature using the Web of Science database, I did not find that anyone was conducted similar investigations. In my opinion the conclusions are correct. References also are OK.

The noticed errors and inaccuracies:

1. Line 151: is” Figure 1a,b”, it should be “Figure 1A,B”.

2. No description of the contents of Figure 1. Most of the text in a description  in Figure 1 (lines 154-165) should be placed before the Figure 1.

3. The above remark can also be applied to Table 1 (lines 173-178).

4. Chapter 4.4: the detailed description of the method must be included, the references leave a lot of ambiguity.

5. Lines 687-706: it is not a method description.

Author Response

We thank the Reviewer 3 for reading the manuscript and their comments.

Reviewer 3

The authors have determined the bimolecular rate constant of scavenging cation radicals of retinyl palmitate, retinaldehyde and retinol by lutein, zeaxanthin, dopa-melanin, ascorbate and α-tocopherol.  The retinoid radical cations were generated by pulse radiolysis of saturated N2O in benzene or Triton X-100 micelles solubilized in phosphate buffer containing retinoids and cation radicals scavengers. Thiocyanate was used as an indicator and transmittance of solutions were measured by transient absorption spectroscopy. Lutein captured free radicals the fastest (8.85-12.6 x 109 M-1s-1) approaching the diffusion-limited limit, the next one was zeaxanthin (6.39-7.9 x 109 M-1s-1). The remaining results were very low, indicating a low rate of cation radical scavenging (<0.002-0.58 x 109 M-1s-1). Based on the conducted research, the authors try to explain the action of the investigated compounds in protection of retinoids involved in vision process. They concluded, among other things, that “While the protection of retinoids by lutein and zeaxanthin appears inefficient in the retinal periphery, it can be quite substantial in the macula”. The topic is original in terms of a new active compound and it address a specific gap in the field. The research and its results are interesting. The discussion of the results is good and refers to many publications. Studying the literature using the Web of Science database, I did not find that anyone was conducted similar investigations. In my opinion the conclusions are correct. References also are OK.

The noticed errors and inaccuracies:

  1. Line 151: is” Figure 1a,b”, it should be “Figure 1A,B”.

Corrected as suggested.

  1. No description of the contents of Figure 1. Most of the text in a description  in Figure 1 (lines 154-165) should be placed before the Figure 1.

We have included all essential information in the Figure 1 caption so the figure can be understandable.

  1. The above remark can also be applied to Table 1 (lines 173-178).

We have included all essential information in the Table 1 caption so it can be understandable.

  1. Chapter 4.4: the detailed description of the method must be included, the references leave a lot of ambiguity.

As suggested, we have expanded the method section.

  1. Lines 687-706: it is not a method description.

We have rephrased that section so it is clear it is a method description.

Round 2

Reviewer 2 Report

Comments and Suggestions for Authors

The manuscript in the present form can be accepted.